# THE VARIATIONAL INFOMAX AUTOENCODER

## ABSTRACT

The Variational AutoEncoder (VAE) can learn simultaneously a representation and a generative model, but often, in order to generate sharper samples, a useless representation is learnt. Starting from the observation that a meaningful representation is an informative one, in this work we analyse the VAE from an information theoretic perspective. We define the concept of capacity for the variational network, and we associate the two tasks of VAE: generation and representation quality, to two information properties of the network: information of the generative model and capacity of the network. Then we suggest that an optimal generative model is the one optimising the Capacity-Constrained InfoMax (CCIM), a theoretical objective learning the maximal informative generative model while maintaining bounded the network capacity. The theoretical assumptions are confirmed by the computational experiments, where between the different families of VAE, the one optimising a variational lower bound of the CCIM generates sharper samples and learns a clustered and robust representation. We call this the Variational InfoMax AutoEncoder (VIMAE).

## 1 INTRODUCTION

A common assumption in machine learning is that any visible data $x \in \mathcal{X}$ is completely described by some generative factor $o$, living in a smaller hidden space $\mathcal{O}$, i.e. $x = g(o)$ with $g$ a (possibly stochastic) generative function. The aim of unsupervised representation learning research is to find a *representation* $z$ of the generative factor $o$ living in a known space $\mathcal{Z}$ describing, as well as $o$, the visible data $x$. This is particularly relevant because the learnt small representation $z$ is task agnostic and, in principle, can be used as input for networks performing different tasks, leading to faster and more robust learning (*generalisation property*), (Rifai et al., 2011).

Many models $f_\phi : \mathcal{X} \to \mathcal{Z}$ trying to learn such representations have been proposed (Dinh et al., 2016; Hinton et al., 2006; Maddison et al., 2017; Radford et al., 2015), but recently in order to solve this problem it was proposed to consider a dual problem: define a priori $z$ and find a generator map $g_\theta$, such that for any $z$, $g_\theta(z)$ is an element of $\mathcal{X}$. In particular, two families of probabilistic generative models have become dominant: Variational AutoEncoder (VAE) (Kingma & Welling, 2013; Rezende et al., 2014) and Generative Adversarial Network (GAN) (Goodfellow et al., 2014). The common idea of the two approaches is that a good generator $p_\theta(x|z)$ is the one able to generate the data that is as close as possible to the visible one, i.e. that with respect a certain metric $D$, the distance between the marginal $p_\theta(x) = \mathbb{E}_{p(z)}[p_\theta(x|z)]$ and the visible distribution $p_D(x)$ is minimal.

In this manuscript we restrict our attention to the VAE model, since by its architecture, it is the only one where the learnt representation, possibly from different datasets, can be used as input for networks performing different tasks, (Achille et al., 2018; Ramapuram et al., 2017). Although VAE, by its training robustness and general good generative performance is the most popular model for representation learning, in particular cases it suffers from the *uninformative representation* issue: the representation is entangled and the generative model tends to be independent of $z$, i.e. $p_\theta(x|z) \approx p_\theta(x)$. As highlighted in the next section such behaviour is intrinsic in the variational loss, the Evidence Lower BOund (ELBO), encouraging a less informative representation.

Following the direction suggested in (Alemi et al., 2017; Zhao et al., 2017) we describe the VAE from an information theoretic perspective. Such description lead us to two observations: the WAE (Tolstikhin et al., 2017) and InfoVAE (Zhao et al., 2017) models are actually maximising a lower bound of the mutual information associated to a generator $p_\theta(x|z)$ belonging in a certain family

$\mathcal{P}$, the Variational InfoMax (VIM); and given that the capacity of the network is a function of the entropy of the prior $p(z)$, the VIM is the variational expression of the Capacity-Constrained InfoMax (CCIM). Following the analogies between the CCIM and the Information Bottleneck (IB), the theoretical principle associated with the ELBO, we deduce that the representation quality is associated to the capacity term.

The theoretical arguments are confirmed by the performed experiments where we observe that, differently from what was argued in previous works (Higgins et al., 2017; Burgess et al., 2018), it is possible to train a model that is able to learn *good* (able to generalise) representations while maintaining optimal generative performance. The main contributions of the paper are summarised in the following points:

- derivation of a variational lower bound for the maximal mutual information of a generaive model belonging in a certain family, see equation 7;

- definition and bounds estimation for the network capacity for a variational autoencoder, see equation 9;

- association of the two main properties of VAE, generation quality and good representation, to two different information concepts, respectively Mutual Information and network capacity;

- proposal of a new learning principle for unsupervised models: the Capacity-Constrained InfoMax, see equation 10, that allows both to learn a good representation while maintaining optimal generative performance.

The work is divided as follows: in the second section we describe briefly the VAE and its variants; in the third and fourth sections we describe the variational infomax method and related work. We conclude the paper with the experimental results and the final observations.

## 2 BACKGROUND

The aim of this section is to describe VAE, understand principal issues of the ELBO objective and describe the three most relevant approaches to overcome such issues.

### 2.1 NOTATION AND PRELIMINARY DEFINITIONS

We use calligraphic letters (i.e. $\mathcal{X}$) for sets, capital letters (i.e. $X$) for random variables, and lower case letters (i.e. $x$) for their samples. With abuse of notation we denote both the probability and the corresponding density with the lower case letters (i.e. $p(x)$).

**KL divergence**   Given two random distributions $p(x)$ and $q(x)$, the Kullback-Leibler (KL) divergence

$$D_{KL}(p(x)||q(x)) = \int \log\left(\frac{p(y)}{q(y)}\right) p(y) dy \tag{1}$$

is an (intuitive) measure of the distance between the distributions $p$ and $q$.

**Mutual Information and Capacity**   Given a channel $Z \to X$ with $X$ and $Z$ random variables, jointly distributed according to $p(x, z)$ and with marginals $p(x)$ and $p(z)$. The mutual information

$$I(X, Z) = D_{KL}(p(x, z)||p(x)p(z)),$$

is a measure of the reduction of uncertainty in $X$ due to the knowledge of $Z$, and the capacity

$$C(X, Z) = \sup_{p(z) \in \mathcal{P}} I(X, Z)$$

is the maximal information that can be shared for a fixed generator $p(x|z)$.

## 2.2 VARIATIONAL AUTOENCODER

From now on let us assume that the unknown distribution of the data $p(x)$ coincides with the empirical one $p_D(x)$, and that the distribution of the latent representation $p(z)$ is known. In this context the VAE is a model solving the following optimisation problem: find the generative model $p_\theta(x, z) \in \mathcal{P}_\theta$, specified by the parameters $\theta$ of the associated neural network, maximising the ELBO objective

$$ELBO_{\theta,\phi} = \mathbb{E}_{p(x)}[-D_{KL}(q_\phi(z|x)||p(z)) + \mathbb{E}_{q(z|x)}[\log p(x|z)]], \qquad (2)$$

a lower bound of the unfeasible-to-compute marginal likelihood $\mathbb{E}_{p(x)}[\log p_\theta(x)]$. The ELBO objective is optimised by a regularised autoencoder, with encoder and decoder parametetrising, respectively, the inference and generative distributions, $q_\phi(z|x)$ and $p_\theta(x|z)$, with $\phi \in \Phi$, $\theta \in \Theta$ and regulariser defined by the *rate* term $D_{KL}(q_\phi(z|x)||p(z))$, measuring the excess number of bits required to encode samples from the encoder using the optimal code designed for $p(z)$.

## 2.3 UNINFORMATIVE REPRESENTATION ISSUE

As underlined in the introduction, the main issue of VAE is that the representations are not really informative of the input data and in the worst case, it is learned a $Z$-independent generative model $p_\theta(x|z) = p_\theta(x)$. Such issues are intrinsic in the ELBO objective, equation 2, that can reach the optimum when $D_{KL}(q_\phi(z|x)||p(z)) = 0$ (Zhao et al., 2017). The latter case means that the representation is completely uninformative, indeed the rate term, which can be rewritten as

$$D_{KL}(q_\phi(z|x)||p(z)) = I_q(X, Z) + D_{KL}(q_\phi(z)||p(z)),$$

is a penalty on the encoding information, and is zero when $I_q(X, Z) = 0$, with $q_\phi(z|x) = q_\phi(z) = p(z)$, i.e. when $q_\phi$ does not encode any information about the input $x$.

We now describe the three most relevant models that try to overcome the uninformative representation issue.

**InfoVAE** In (Zhao et al., 2017) the InfoVAE family of models was proposed, a generalisation of the VAE model optimising the objective

$$-\alpha I_q(X, Z) - \lambda D_{KL}(q_\phi(z)||p(z)) + \mathbb{E}_{p(x)}[\mathbb{E}_{q(z|x)}[\log p(x|z)]],$$

with $\alpha$ and $\lambda$ two real positive hyper-parameters.

The main advantage of this definition is that it is possible to consider separately the two components of the rate term. In particular, in (Zhao et al., 2017) it was observed that by eliminating the information penalty ($\alpha = 0$), the generative performance of the model improves and the representation results are more informative.

**$\beta$-VAE** In (Higgins et al., 2017), starting from the observation that the optimal case is rare, but most of the learned features by VAE are not disentangled, it is proposed an opposite approach: put a high penalty to the rate term, in order to constrain the model to learn the most informative property of the data, and then have a disentangled representation of the data. The $\beta$-VAE family is a particular case of InfoVAE where $\alpha = \lambda \gg 1$. This idea, that at first sight looks counter-intuitive, is based on the observation that by the additive property of the KL-divergence

$$D_{KL}(q_\phi(z|x)||p(z)) = \sum_{i=1}^{dim(\mathcal{Z})} D_{KL}(q_\phi(z_i|x)||p(z_i)) \qquad (3)$$

pushing the penalty associated with the rate is equivalent to penalising the informativeness of most features, leaving few features containing the relevant information. Starting from a bits-back coding argument, a similar conclusion was derived in (Chen et al., 2016).

**Minimal rate bound** In order to avoid the null rate issue, and maintain semantic informative representation, in (Alemi et al., 2017) was suggested to optimise the following variant of ELBO:

$$\mathbb{E}_{p(x)}[\mathbb{E}_{q(z|x)}[\log p(x|z)]] - \beta|R_0 - D_{KL}(q_\phi(z|x)||p(z))|, \qquad (4)$$

where $R_0 > 0$, defines both the lower bound of the rate and the upper bound of the encoding information $I_q(X, Z)$. If, as we will see below, the objective in equation 4 is a generalisation of both InfoVAE ($R_0 = \max_\theta I_\theta(X, Z)$) and $\beta$-VAE ($R_0 = 0$), the correct $R_0$ is unknown and its optimisitation is not an easy task.

## 3 THE MODEL

### 3.1 THE VARIATIONAL INFOMAX

Assuming the distribution associated to the two random variables $p(x)$ and $p(z)$ is known, the InfoMax objective is defined as: find the joint distribution $p_\theta(x, z) \in \mathcal{P}_\theta := \{p_\theta(x, z) : \mathbb{E}_{p(z)}[p_\theta(x|z)] = p(x), \quad \mathbb{E}_{p(x)}[p_\theta(z|x)] = p(z)\}$ maximising the mutual information $I_\theta(X, Z) = D_{KL}(p_\theta(x, z)||p(x)p(z))$, i.e. find $\theta^* \in \Theta$ s.t. $I_{\theta^*} \geq I_\theta$ for any $\theta \in \Theta$.

Since the definition via KL divergence is computationally intractable, it is necessary to re-write the mutual information as

$$I_\theta(X, Z) = h_\theta(X) - h_\theta(X|Z), \tag{5}$$

where $h_\theta(X) = -\mathbb{E}_{p_\theta(x)}[\log p_\theta(x)]$ is the entropy of $X$, and $h_\theta(X|Z) = -\mathbb{E}_{p_\theta(x,z)}[\log p_\theta(x|z)]$ is the conditional entropy $h_\theta(X|Z)$. Since $p_\theta(x, z) \in \mathcal{P}_\theta$ the entropy $h_\theta(X) = h(X)$ is constant, and in order to maximise the mutual information it is sufficient to minimise the conditional entropy.

Excluding some special cases (Bell & Sejnowski, 1997), minimising the conditional entropy is unfeasible, so it is necessary to consider an associated variational problem: for any $q_\phi(z|x)$ such that $q_\phi(z) = p(z)$ and $\phi \in \Phi = \Theta$, learn the generative model $p_\theta(x|z)$ minimising the reconstruction accuracy term $\mathbb{E}_{p(x)}[\mathbb{E}_{q_\phi(z|x)}[\log(p_\theta(x|z))]]$. Indeed, the following variational objective:

$$I_{\theta,\phi}(X, Z) = h(X) + \mathbb{E}_{p(x)}\mathbb{E}_{q(z|x)}[\log p_\theta(x|z)] \qquad \text{s.t. } q_\phi(z) = p(z) \tag{6}$$

is a lower bound of $I_{\theta^*}(X, Z)$ and is maximal when $q(z|x) = p_\theta(z|x) = p_{\theta^*}(z|x)$, (see the Appendix).

Unfortunately, the formulation in equation 6 is still unfeasible to compute, because it requires that $q_\phi(z) = p(z)$, but by the butterfly architecture of the autoencoder, $q_\phi(z)$ tends to be uniformly distributed on the space $\mathcal{Z}$. For this reason, the model is trained maximising the following relaxed form:

$$VIM_{\theta,\phi} = \mathbb{E}_{p(x)}\mathbb{E}_{q(z|x)}[\log p_\theta(x|z)] - \lambda D(q_\phi(z)||p(z)), \tag{7}$$

where it is introduced a term $D(q_\phi(z)||p(z))$ encouraging the empirical distribution $q_\phi(z)$ to be close, according to the metric $D$, to $p(z)$. In the following we assume $D = D_{KL}$, and in order to avoid any confusion the variational autoencoder trained maximising equation 7 will be dubbed VIMAE.

**Encoding channel**   In VAE we observed that an uninformative representation was caused by the non-informativeness of the encoding map $q_\phi(z|x)$. Since from equation 7 it is not clear how $q_\phi(z|x)$ behaves, we consider an equivalent representation, (see the Appendix):

$$VIM_{\theta,\phi} = -D_{KL}(p(x)||p_\theta(x)) - (\lambda - 1)D_{KL}(q_\phi(z)||p(z)) + I_{\theta,\phi}(X, Z). \tag{8}$$

From equation 8 we see that the infomax objective, equation 7, can be read as a composition of three sub-objectives: find a generative model $p_\theta(x|z)$, with marginal resembling the visible distribution $p(x)$ (first term); maximise the (unbounded) variational mutual information (third term); and learn an inferred distribution $q_\phi(z, x)$ close to the generative model $p_\theta(x, z)$. Then the optimum is obtained by $q_\phi(x, z) = p_\theta(x, z)$ such that $I_\theta(X, Z)$ is maximal, confirming the validity of the approximation made above.

### 3.2 CHANNEL CAPACITY

In a channel with variational mutual information $I_{\theta,\phi}$ as defined in equation 6 the (variational) capacity $C_{\theta,\phi}(X, Z)$, is defined as

$$C_{\theta,\phi}(X, Z) = \sup_{\theta,\phi,p(z)\in\mathcal{P}} I_{\theta,\phi}(X, Z). \tag{9}$$

If $\mathcal{P}$ is the space of all distributions on $\mathcal{Z}$, the capacity of the network coincides with the variational mutual information of the model trained minimising only the reconstruction loss. In the latter case, as observed above, it is not guaranteed to learn the generator with $p_\theta(z) = p(z)$. This is because, given two equally informative generative models $p_\theta(x|z)$ and $p_{\theta'}(x|z)$, with encoder respectively $q_\phi(z|x)$ and $q_{\phi'}(z|x)$, with $h(q_\phi(z)) < h(q_{\phi'}(z))$, then $I_{\theta,\phi} < I_{\theta',\phi'}$. From such observation we deduce that bounding the entropy of the representation $Z$ is the way to bound the capacity without penalising the encoding information, and that the penalty introduced in equation 7 is actually a bound to the capacity itself. So, the VIM objective can be defined as a variational approximation of the Capacity-Constrained InfoMax:

$$\max_{Z \in \mathcal{Z}} I(X, Z) - \lambda C(X, Z), \tag{10}$$

which given a set of equally informative generators, learns the one having the minimal capacity. The idea of the CCIM is similar to the idea of the IB, from which was derived the $\beta$-VAE: constrain the capacity of the network in order to learn only the relevant features of the input data. The difference with the IB,

$$\max_{Z \in \mathcal{Z}} I(X, Z) - \lambda I_q(X, Z) \tag{11}$$

lies in the second term: the network capacity instead of the encoding information. This choice allows the network to learn a good representation (small capacity) while maintaining good generative performance (high mutual information). Indeed, as shown in equation 8, the generative performance is associated to the informativeness of $q_\phi(z|x)$ and $p_\theta(x|z)$.

In order to test the assumption that it is sufficient to bound the entropy of $Z$, instead of the encoding mutual information, to learn a good representation, in the experiments (see below) we consider the cases $Z$ is Normal (VIMAE-n) or Logistic (VIMAE-l) distributed. We choose to compare the popular Normal distribution with the Logistic one for two reasons: the Logistic has less entropy than a Gaussian distribution and because it is a common assumption in natural science to suppose that the hidden factors of the visible data are logistically distributed (Hyvärinen et al., 2009).

**Entropy vs Rate**   According to what written above the choice of the distribution $p(z)$ is a way to bound the maximal information $I_\theta$ of the generative model as well as $R_0$ in equation 4, indeed, if we choose $R_0 = C(X, Z)$ the theoretical solutions of equation 4 and equation 7 coincide. Although the two approaches are similar, both defines a target $I_\theta$ for the generator $p_\theta(x|z)$, they differ in one fundamental aspect: the ratio between the information $I_\theta$ and network capacity $C$. For any given choice of the prior $p(z)$ the generative model maximising VIM is the one with $I_\theta = C$, instead for any $R_0 < C$ the information $I_\theta$ associated to the generative model learned, by equation 4, coincides with $R_0$, i.e. $I_\theta = R_0 < C$. That implies an high sensitivity to the hyper-parameter $R_0$ for the model in equation 4. In fact, as suggested in (Mathieu et al., 2018), a ratio $I_\theta/C \approx 1$, guarantees an appropriate level of overlap in the latent space between latent encoding, and then a better representation quality.

## 4   RELATED WORK

**Autoencoder literature**   Autoencoder models are one of the most used family of neural networks to extract features in an unsupervised way (Bengio et al., 2013), and their relationship with Information Theory is well-established from the first unregularised autoencoders (Baldi & Hornik, 1989). The classical unregularised autoencoders, minimising the reconstruction loss $\mathbb{E}_{p(x)}[\mathbb{E}_{q_\phi(z|x)}[-\log p_\theta(x|z)]]$, are maximising an unbounded information, i.e. they are looking for a solution in the space $\tilde{\mathcal{P}}_\theta = \{p_\theta : p_\theta(x) = p(x)\}$. In general, a solution in this wide space is good only for reconstruction performance because $Z$ contains all the possible information that can be stored in the space $\mathcal{Z}$, and is not robust to input noise (Vincent et al., 2008); but, as observed in (Grover & Ermon, 2018), if $q_\phi(z|x) \sim \mathcal{N}(\mu(x), \sigma(x))$ the model $p_\theta(x|z)$ is robust to noise and is a Gaussian generator. In this context the uninformative issue is avoided, but the price to pay is the impossibility to sample directly from a prior $p(z)$ that is not defined; indeed, the model described in (Grover & Ermon, 2018) requires running relatively expensive Markov Chain to obtain samples.

Many regularised models have been proposed, but the most well known is VAE, that minimises the expected code length of communicating $x$. As we observed in the previous sections, it is not

guaranteed that the method finds a useful representation. Such issue can be solved both controlling the information of the model, or considering a more flexible prior $p(z)$,(Rezende & Mohamed, 2015; Kingma et al., 2016; Dinh et al., 2016). In this manuscript we do not consider the latter approach, because by the model structure, it requires quite expensive computation (autoregressive models are necessary) and because the learned representation is not suitable for task different than the generation such as lifelong context, for which the standard VAE is used with success (Achille et al., 2018).

The objective in equation 7 was firstly derived in (Tolstikhin et al., 2017) and (Zhao et al., 2017). Particularly relevant is the derivation in (Tolstikhin et al., 2017) because it allows us to describe an informative model $p_\theta(x, z)$ as the one minimising the transport cost between the original and generated data.

Finally, we underline that in case we wish to consider a Jensen-Shannon divergence in equation 7 it is necessary to consider an adversarial network model, discriminating the true samples $z \sim p(z)$ from the fake sampled by $q_\phi(z)$ (Goodfellow et al., 2014). In the latter case the obtained model is equivalent to the Adversarial AutoEncoder (Makhzani et al., 2015). We conclude by remarking that in all the cases cited above the Infomax objective was never maximised using a prior $p(z)$ different from a Gaussian.

**Information theoretic literature** Information theory is strongly related with neural networks, and not only with autoencoders. Originally the InfoMax objective was applied to a self-organised system with a single hidden layer, (Bell & Sejnowski, 1997; Linsker, 1989) where the bound in the capacity was given by the numbers of hidden neurons. More recently, the (naive) InfoMax has given way to a new information-theoretic principle: the Information-Bottleneck (Tishby et al., 2000). The idea of this principle is that a feed-forward neural network trained for task $T$ tends to learn a minimal sufficient representation of the data, maximising the following objective:

$$\max_Z I(Z, T) - \beta I(X, Z). \tag{12}$$

Although it was shown that in the general case this principle does not hold true (Saxe et al., 2018), the principle was used as a regularisation technique with success both in unsupervised (Alemi et al., 2017; Higgins et al., 2017) and supervised (Alemi et al., 2016) settings. We observe that the VIM, equation 7, and IB, equation 12, coincide in the case of a deterministic encoder, where the encoding information is the entropy of $Z$.

## 5 EXPERIMENTS

Here we empirically evaluate the VIMAE. The section is divided into three parts: in the first part we compare the ability of the described models to infer the representation, $z \sim \mathcal{N}(0, I)$. Such an experiment is to evaluate the entropy of $Z \sim q(Z)$ and then, as observed in section 3.2, an indirect way to estimate capacity of the network ($C(X, Z) \propto h(Z)$). In the second part we evaluate the reconstruction and generative performance of the models. Indeed, the combination of the two tasks is estimation of the mutual information of the generative model $I_\theta(X, Z)$, see equation 8.

In the third part we evaluate the robustness to noise and generalisation property of the learnt representation, observing that an informative model with small capacity is the best one for these tasks.

In all the described experiments, the divergence $D_{KL}(q(z)||p(z))$ in equation 7 is approximated via the Maximum Mean Discrepancy (Zhao et al., 2017) defined as:

$$\text{MMD}(q(z), p(z)) = \sup_{f:\|f\|_{\mathcal{H}_k} \leq 1} \mathbb{E}_{p(z)}[f(Z)] - \mathbb{E}_{q(z)}[f(Z)] \tag{13}$$

where $\mathcal{H}_k$ is the Reproducing Kernel Hilbert Space associated to a positive definite kernel $k(\cdot, \cdot)$ : $\mathcal{Z} \times \mathcal{Z} \to \mathbb{R}_+$. Moreover, by difficulties to compute the objective equation 4, as suggested in (Alemi et al., 2017) we decided to optimise a $\beta$-VAE, denoted $\beta_A$-VAE, with $\beta < 1$; in order to avoid any confusion, the original version proposed in (Higgins et al., 2017) with $\beta \gg 1$ will be renamed $\beta_H$-VAE.

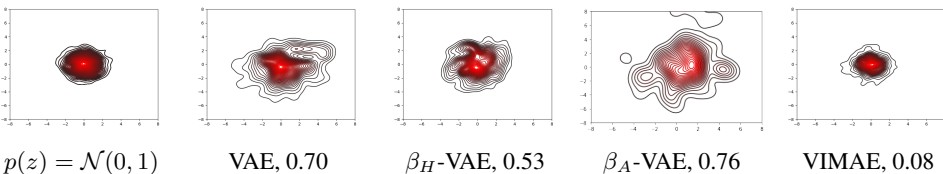

$p(z) = \mathcal{N}(0,1)$     VAE, 0.70     $\beta_H$-VAE, 0.53     $\beta_A$-VAE, 0.76     VIMAE, 0.08

Figure 1: 2-d learned representations. Under each plot the model name is followed by the respective MMD value.

## 5.1 THE ENTROPY OF $Z$

Experiments in this part are performed with an autoencoder trained with the MNIST data-set, a collection of 70k monocromatic handwritten digits, where both the inference and generative distribution are modelled by 3-layer deep neural nets with 256 hidden units in each layer and $\mathcal{Z} = \mathbb{R}^2$.

From the 2d hidden distribution learnt by the different methods in figure 1, we observe that the ELBO-based models are not able to learn a distribution $q(z)$ that is close to the prior $p(z)$; and, as we will see in the following experiments, $\beta_H$-VAE, the only ELBO-model having comparing results to the VIMAE, has to penalise drastically the decoding information.

Thanks to this experiment we see that, in general, the ELBO-based models are not seeking for a small capacity network, since $h(q(z)) \gg h(p(z))$, and that the divergence penalty introduced in equation 7 is a bound for the entropy $h(q(z))$, and then of the channel capacity $C$.

## 5.2 THE MODEL INFORMATION

The experiments in these final sections were performed with the same settings and autoencoder models used in (Tolstikhin et al., 2017), an architecture similar to the DCGAN (Radford et al., 2015) with batch normalization (Ioffe & Szegedy, 2015) (more details given in the Appendix). We consider four data-sets: MNIST and CIFAR10, two standard data-sets with ground-truth labels; Omniglot, a data-set of 1623 characters from 50 alphabets, 30 training and 20 evaluation, where each character appears 80 times, to evaluate the informativeness of the model and the quality of the learned representation; in the Appendix, we also consider CelebA (Liu et al., 2015), consisting of roughly of 203k faces of $64 \times 64$ resolution, in order to compare the generative quality of the pictures. After considering many parameters for $\beta_H$, $\beta_A$ and $\lambda$, we choose, in accordance with what was suggested in (Tolstikhin et al., 2017), $\beta_H = \lambda = 10$, and $\beta_A = 0.2$ for MNIST and Omniglot and $\beta = \lambda = 100$ and $\beta_A = 0.4$ for CelebA and CIFAR10 experiments.

The goal of this section is to evaluate the informativeness of the learnt generative model $p_\theta(x|z)$. From what was described above, the reconstruction loss or the generation quality alone are not reliable metrics, because the reconstruction loss is an estimation of the variational mutual information $I_{\theta,\phi}(X,Z)$, and the generation quality is an estimation of $D_{KL}(p_\theta(x)||p(x))$ that, as observed in section 2.3, it is possible to minimise with an uninformative generator. But, according to equation 8 the combination of the two task performances is a good empirical estimation of $I_\theta(X,Z)$; indeed, by generative experiments, we require that $q(z) = p(z)$, so the KL divergence term in equation 8 comes for free.

**Reconstruction and generative performances** According to what was asserted above, we observe from figure 2 (top) and figure 3 that an high penalty on the rate term (small encoding information) in the ELBO based model coincides with poor reconstruction. Indeed, the $\beta_H$-VAE, is the one with worst reconstruction, and instead $\beta_A$-VAE is the model that behaves similarly to VIMAE, that is optimising an encoding information-free objective. Such behaviour is consistent with the rate term associated to each model, see caption of each figure, where we observe that VIMAE, have the highest rate and in particular this is twice the one associated to $\beta_A$-VAE, confirming that VIMAE learns a maximal informative decoder, and that $\beta_A$-VAE is only theoretically analogous to VIMAE.

The models that we are considering are defined as generative models, so giving a sample $z \sim p(z)$ they should be able to generate a new data $x$ similar to the original one. As we observe from the

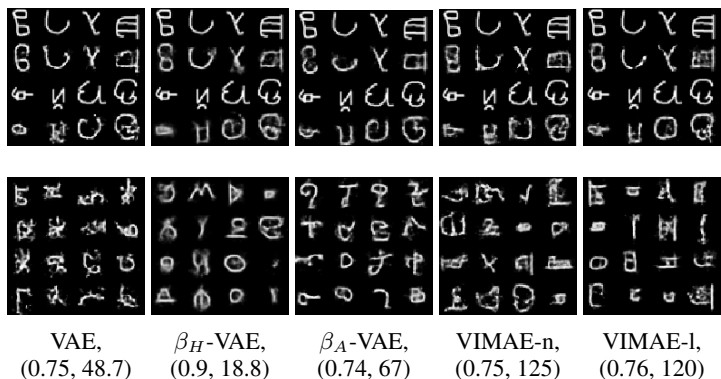

| VAE, | $\beta_H$-VAE, | $\beta_A$-VAE, | VIMAE-n, | VIMAE-l, |
|---|---|---|---|---|
| (0.75, 48.7) | (0.9, 18.8) | (0.74, 67) | (0.75, 125) | (0.76, 120) |

Figure 2: Test reconstruction (top) and random generative samples (bottom) of the different methods with Omniglot (right). In test reconstructions, the odd rows are the original data. Each model is denoted by its name and the reconstruction $\| \cdot \|_2$ and rate term.

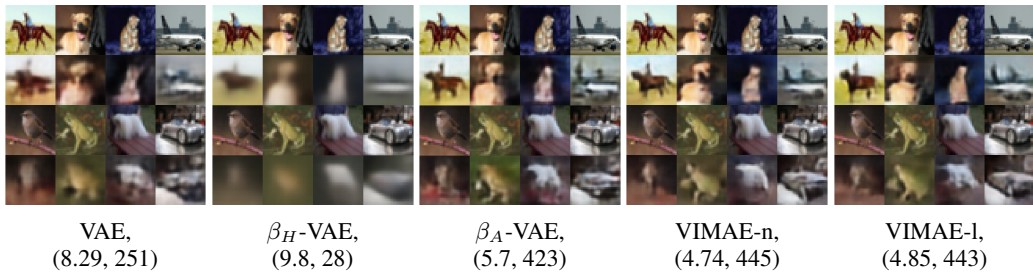

| VAE, | $\beta_H$-VAE, | $\beta_A$-VAE, | VIMAE-n, | VIMAE-l, |
|---|---|---|---|---|
| (8.29, 251) | (9.8, 28) | (5.7, 423) | (4.74, 445) | (4.85, 443) |

Figure 3: Test reconstruction, CIFAR 10. Odd rows are the original data. Each model is denoted by its name and the reconstruction $\| \cdot \|_2$ and rate term.

generated samples of Omniglot in figure 2 (bottom) and Negative LogLikelihood (NLL) listed in table 1, both VAE and $\beta_H$-VAE do not generate good samples, confirming that sharper samples are associated to informative generators. Such qualities on Omniglot are confirmed by the experiments with the CIFAR10 and CelebA data-set (Liu et al., 2015), see the Appendix, where it is observed that if the generative difference between the two VIMAEs is small, the difference with the VAE counterparts is high. Such behaviour is in agreement with what was observed until now: the ELBO based model does not learn a good generative network, and the good reconstruction is simply associated to a large entropy of $Z$, (encoding information) instead of an informative generative model $p_\theta(x|z)$.

Similar experiments were conducted with the MNIST dataset and are discussed in the appendix.

### 5.3 GENERALISATION PROPERTY

We defined a good representation as the one containing the relevant properties of the visible data and able to generalise from the task for which was trained. In order to evaluate such quality, following the approach proposed in (Rifai et al., 2011), we evaluate the accuracy of an SVM directly trained on the learned features of the data. Proceeding as in (Zhao et al., 2017), we train the M1+TSVM (Kingma et al., 2014) and use the semi-supervised performance over 1000 (100 for Omniglot) samples as an approximate metric to verify the relevance and the quality of the learned representation. In order to evaluate the robustness of the learned features, we performed the same algorithm on the representation associated to corrupted data, i.e. $z \sim q(z|x+\nu)$, considering two types of noise: Gaussian and mask. In the Gaussian case, we add to each pixel a $\nu$ value sampled from $\mathcal{N}(0, \sigma^2)$ with $\sigma \in \{0.2, 0.3, 0.4\}$, and in the masking

Table 1: NLL for generated samples on Omniglot (smaler is better)

| Method | NLL |
|---|---|
| VAE | 1224 |
| $\beta_H$-VAE | 1254 |
| $\beta_A$-VAE | 1228 |
| VIMAE-n | **1190** |
| VIMAE-l | 1223 |

Table 2: Semi-supervised classification CIFAR10.

| Method | accuracy (%) | | |
|---|---|---|---|
| | $\nu = 0$ | $\mathcal{N}(0, 0.3^2)$ | $\mathcal{B}(0.2)$ |
| VAE | 30 | 25 | 16 |
| $\beta_H$-VAE | 29 | 26 | 19 |
| $\beta_A$-VAE | 31 | 31 | 18 |
| VIMAE-n | 29 | 28 | **23** |
| VIMAE-l | **32** | **34** | **23** |

Table 3: Semi-supervised classification, MNIST.

Table 4: Semi-supervised classification, Omniglot (random sampling: 20%).

| Method | accuracy (%) | | | | | accuracy (%) | | | | |
|---|---|---|---|---|---|---|---|---|---|---|
| | $\nu = 0$ | $\nu = \mathcal{N}(0, \sigma^2)$ | | $\nu = \mathcal{B}(p)$ | | $\nu = 0$ | $\nu = \mathcal{N}(0, \sigma^2)$ | | $\nu = \mathcal{B}(p)$ | |
| | | 0.2 | 0.4 | 0.2 | 0.5 | | 0.2 | 0.4 | 0.2 | 0.5 |
| VAE | 80 | 77 | 70 | 72 | 52 | 22 | 22 | 17 | 22 | 16 |
| $\beta_H$-VAE | 92 | 86 | 82 | 91 | 84 | 21 | 21 | 22 | 19 | 17 |
| $\beta_A$-VAE | **93** | 66 | 13 | 85 | 65 | 22 | 22 | 21 | 21 | 24 |
| VIMAE-n | **93** | **92** | 86 | **92** | 86 | 22 | **23** | **24** | 22 | **22** |
| VIMAE-l | **93** | **92** | **88** | **92** | **87** | **24** | **23** | 20 | **23** | **22** |

case a fraction $\nu$ of the elements is forced to be 0: each pixel is masked according to a Bernoulli distribution $\mathcal{B}(p), p \in \{0.2, 0.5\}$. Higher classification performance suggests that the learned representation contains the relevant information and, in case of corrupted input data, that it is robust. In the Omniglot case by the challenge of the task (the test alphabet was never seen in the training) we consider a 5-character data-set, split into 300 ($60 \times 5$) for training and 100 for evaluation.

From the classification scores listed in tables 2- 4, we see that the ELBO-based model learnt good representations for clean data, but not when corrupted data is given as input. This is particularly clear in the Bernoulli case, that is a noise different from the one seen in the training. Particularly relevant is the behaviour of the two VIMAEs: they are comparable in the cases of clean data and small noise, but the one with big capacity, VIMAE-n, suffers in large noise, while the one with small capacity, VIMAE-l, is the most robust and in some challenging cases, see table 2, the noise helps to improve the model accuracy. Such a result is consistent with the idea that a small capacity network is learning the relevant factors of the input data, that are the only ones robust to the input noise.

## 6 CONCLUSION

We observe, via an information theoretic description of VAE, that it is possible to learn a good generative model while maintaining a meaningful hidden representation, and that goal can be reached by optimising the CCIM, an objective that separates out the two properties of a network: the generative information and its capacity. We compare both theoretically and computationally the CCIM objective with the IB and its variation (Alemi et al., 2017), observing that the CCIM optimal solution, in particular cases, coincides with the IB one, but that in the practical case the difference is substantial.

Future work includes the extension of the VIMAE model to a lifelong task, a scenario where it is necessary to learn both an informative representation while maintaining bounded the capacity of the network, in order to avoid the catastrophic forgetting issue (Achille et al., 2018).

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

### RELATIONSHIP BETWEEN ENCODING, DECODING AND VARIATIONAL INFORMATION

Defined $q(z, x) := q_\phi(z|x)p(x)$ the encoding distribution and $p_\theta(x, z) := p_\theta(x|z)p(z)$ the decoding one. The encoding $I_q(X, Z)$, decoding $I_\theta(X, Z)$ and the variational $I_{\theta,\phi}(X, Z)$ information are defined respectively:

$$I_q(X, Z) = D_{KL}(q(z, x)||q(z)p(x)) = h(X) - h_q(X|Z)$$
$$I_\theta(X, Z) = D_{KL}(p_\theta(z, x)||p(z)p(x)) = h(X) - h_\theta(X|Z)$$
$$I_{\theta,\phi}(X, Z) = h(X) - \mathbb{E}_{q(z,x)}[-\log p_\theta(x|z)]$$

Assuming $\theta^* \in \Theta$ is the parameter associated to the maximal decoding information, $I_\theta(X, Z) \leq I_{\theta^*}(X, Z)$ for any $\theta \in \Theta$, it follows that for any $q_\phi(x, z) \in \mathcal{P}_\theta$, i.e. for any $q_\phi(z) = \mathbb{E}_{p(x)}[q_\phi(x|z)] = p(z)$ and $\phi \in \Phi \subset \Theta$,

$$I_{\theta^*}(X, Z) \geq I_q(X, Z).$$

Then a lower bound of $I_q$ is a lower bound of $I_{\theta^*}$. By property of KL-divergence we have that for any $p_\theta(x|z)$ the following relationship holds:

$$\mathbb{E}_{q(z,x)}[-\log p_\theta(x|z)] = h_q(X|Z) + \mathbb{E}_{q(z)}[D_{KL}(q_\phi(x|z)||p_\theta(x|z))] \tag{14}$$

From equation 14 and the definition of the variational information $I_{\theta,\phi}$ we deduce that:

$$I_{\theta^*}(X, Z) \geq I_q(X, Z) \geq I_{\theta,\phi}(X, Z)$$

We conclude observing that if $\Theta = \Phi$, at optimum the three information terms above are equal, and then $q_\phi(x, z) = p_\theta(x, z)$.

*Observation*: If $\Phi \subset \Theta$, the variational mutual information can be at most equal to $I_{q^*}$, the maximal $I_q$, but it is not guaranteed that $I_{q^*} = I_{\theta^*}$.

### DERIVATION OF EQUATION 8

In (Zhao et al., 2017), it is observed that equation 7 can be written as follows:

$$VIM_{\theta,\phi} = -D_{KL}(p(x)||p_\theta(x)) - \mathbb{E}_{p(x)}[D_{KL}(q_\phi(z|x)||p_\theta(z|x))]-$$
$$- (\lambda - 1)D_{KL}(q_\phi(z)||p(z)) + I_q(X, Z).$$

From equation above, to verify that equation 8 is correct, it is sufficient to show that

$$I_{\theta,\phi}(X, Z) = I_q(X, Z) - \mathbb{E}_{p(x)}[D_{KL}(q_\phi(z|x)||p_\theta(z|x))]. \tag{15}$$

The equation 15 follows by the property of the autoencoder and equation 15.

More precisely, by equation 14 we have that

$$I_{\theta,\phi}(X, Z) = I_q(X, Z) - \mathbb{E}_{q(z)}D_{KL}(q_\phi(x|z)||p_\theta(x|z))$$

and, by AutoEncoder architecture, as observed in section 3, $p_\theta(z) = \mathbb{E}_{p(x)}[p_\theta(z|x)] = q(z)$. Then the following equation holds

$$\mathbb{E}_{q(z)}D_{KL}(q_\phi(x|z)||p_\theta(x|z)) = \mathbb{E}_{p(x)}D_{KL}(q_\phi(z|x)||p_\theta(z|x)).$$

Indeed,

$$\mathbb{E}_{q(z)}D_{KL}(q_\phi(x|z)||p_\theta(x|z)) = \int q(z) \int q(x|z) \log \frac{q(x|z)}{p_\theta(x|z)}dxdz$$
$$= \int \int q(x)q(z|x) \log \frac{q(x|z)q(z)}{p_\theta(x|z)(z)}dxdz = \int p(x) \int q(z|x) \log \frac{q(z|x)}{p_\theta(z|x)}dxdz =$$
$$= \mathbb{E}_{p(x)}D_{KL}(q_\phi(z|x)||p_\theta(z|x)).$$

## FURTHER DETAILS ON EXPERIMENTS

In all the experiments in section 5.2 we considered the latent space $\mathcal{Z} = \mathbb{R}^d$, for all the models we choose the prior $p(z)$ to be a Gaussian with zero mean and identity covariance, only in VIMAE-l we choose the prior $p(z)$ to be a logistic with mean zero and identity variance. We choose $p_\theta(x|z)$ to be similar to DCGAN with batch normalization and $q_\phi(z|x)$ to be a convolutional deep neural network. The entire models are trained end to end by Adam (Kingma & Ba, 2014) with $\alpha = 10^{-3}, \beta_1 = 0.5, \beta_2 = 0.999$. We considered a deterministic decoder and we approximate the reconstruction loss with the $L_2$ loss, i.e. $\mathbb{E}_{p(x)}[\mathbb{E}_{q(z|x)}[-\log p_\theta(x|z)]] = \|x - x_g\|_2^2$, with $x_g$ indicating the generated datum. In VIMAE case, while training we were adding a pixel-wise Gaussian noise truncated at 0.01 to all the images before feeding them to encoder, in order to make the encoder random. In VAE and $\beta-$VAE case, instead we used the standard reparameterization trick (Kingma & Welling, 2013).

In the following we describe the data-sets considered and the associated neural networks, we follow the same description given in (Tolstikhin et al., 2017) since we used the same neural nets.

### MNIST AND OMNIGLOT

MNIST is a data-set containing 70k grey-scale handwritten digits and associated labels of resolution $28 \times 28$, subdivided in three subsets: train (50k), validation (10k) and test (10k).
Omniglot is a data-set containing 1623 different handwritten characters of resolution $28 \times 28$ from 50 different alphabets. Each of the 1623 characters was drawn online via Amazon's Mechanical Turk by 20 different people. In the same fashion as done in (Vinyals et al., 2016) we considered an augmented version where each character is rotated respectively by 90, 180, 270 degrees, in this way each character appears 80 times. The Omniglot data-set although has the same resolution of the MNIST, for this reason we use the same network, it is more challenging because, it is more entropic then MNIST, in fact the classes move from 10 to 1623 and the test classes are never seen in the training.

We choose $\mathcal{Z} = \mathbb{R}^8$, and $\beta = \lambda = 10$, we used mini-batches of size 100 and trained the model for 80 epochs. Both encoder and decoder used fully convolutional architectures with $4 \times 4$ convolutional filters.
Encoder:

$$x \in \mathbb{R}^{28 \times 28} \to \text{Conv}_{128} \to \text{BN} \to \text{ReLu}$$
$$\to \text{Conv}_{256} \to \text{BN} \to \text{ReLu}$$
$$\to \text{Conv}_{512} \to \text{BN} \to \text{ReLu}$$
$$\to \text{Conv}_{1024} \to \text{BN} \to \text{ReLu}$$

Decoder:

$$z \in \mathbb{R}^8 \to \text{FC}_{7 \times 7 \times 1024}$$
$$\to \text{FSConv}_{512} \to \text{BN} \to \text{ReLu}$$
$$\to \text{FSConv}_{256} \to \text{BN} \to \text{ReLu} \to \text{FSConv}_1$$

Where $\text{Conv}_k$ stands for a convolution with $k$ filters, $\text{FSConv}_k$ for the fractional strided convolution with $k$ filters, BN for batch normalization, ReLU for the rectified linear units, and $\text{FC}_k$ for the fully connected layer mapping to $\mathbb{R}^k$. All the convolutions in the encoder used vertical and horizontal strides 2 and SAME padding.

### CELEBA AND CIFAR10

CelebA is a data-set with 202 599 faces images. We preprocessed the images by first taking a $140 \times 140$ center crops and then resizing to the $64 \times 64$ resolution and we consider the last 20k images as test subset.
CIFAR10 is a dataset consisting of of 60k $32 \times 32$ colour images in 10 classes, with 6k images per class. There are 50k training images and 10k test images.

For these data-sets we choose the same network with the same hyper-parameters, $\lambda = 100$ and $\mathcal{Z} = \mathbb{R}^{64}$. We used mini-batches of size 100 and trained the model for 60 epochs. Both encoder and decoder used a fully convolutional architectures with $5 \times 5$ convolutioanal filters.
Encoder:

$$x \in \mathbb{R}^{64 \times 64 \times 3} \rightarrow \text{Conv}_{128} \rightarrow \text{BN} \rightarrow \text{ReLu}$$
$$\rightarrow \text{Conv}_{256} \rightarrow \text{BN} \rightarrow \text{ReLu}$$
$$\rightarrow \text{Conv}_{512} \rightarrow \text{BN} \rightarrow \text{ReLu}$$
$$\rightarrow \text{Conv}_{1024} \rightarrow \text{BN} \rightarrow \text{ReLu}$$

Decoder:

$$z \in \mathbb{R}^{64} \rightarrow \text{FC}_{8 \times 8 \times 1024}$$
$$\rightarrow \text{FSConv}_{512} \rightarrow \text{BN} \rightarrow \text{ReLu}$$
$$\rightarrow \text{FSConv}_{256} \rightarrow \text{BN} \rightarrow \text{ReLu}$$
$$\rightarrow \text{FSConv}_{128} \rightarrow \text{BN} \rightarrow \text{ReLu} \rightarrow \text{FSConv}_1$$

## CELEBA AND OMNIGLOT EXPERIMENTS

In section 5.2 we evaluate the generative performance on relatively simple grey-scale data-set. In order to consider a more challenging data-set and quantitatively compare the generative performances of the trained models we evaluate the Frechet Inception Distance (FID) on CelebA and CIFAR10 based on $10^4$ samples. From table 5, we observe, in agreement on what observed in section 5.2 and figures 4 and 5 that the difference between the two VIMAE models is minimal, instead it is big the difference with the ELBO based models ($\beta_H$-VAE is not listed in table 5, because it does not converge).

Table 5: FID scores for generated samples on CelebA (smaller is better)

| Method | FID | |
|---|---|---|
| | CelebA | CIFAR10 |
| VAE | 82 | 168 |
| $\beta_H$-VAE | - | 262 |
| $\beta_A$-VAE | 89 | 174 |
| VIMAE-l | 56 | **103** |
| VIMAE-n | **55** | 104 |

## MNIST EXPERIMENTS

The MNIST dataset is a classical benchmark but quite simple, indeed as we see from figure 6 all the models considered are able to reconstruct without big differences, although as we see from the $L_2$ reconstruction loss in the VIMAE models is smaller than the ELBO-based counterpart. That results combined with the high value of the rate term, suggest that the VIMAE are actually learning a maximal informative decoder. We conclude observing that, according to what seen until now, the high rate value for $\beta_A$-VAE is associated to an high entropy $h(q(z))$, this is well visible by the generated samples from $p(z)$ in figure 7, where the samples associated to $\beta_A$-VAE seem to belong to the same cluster.

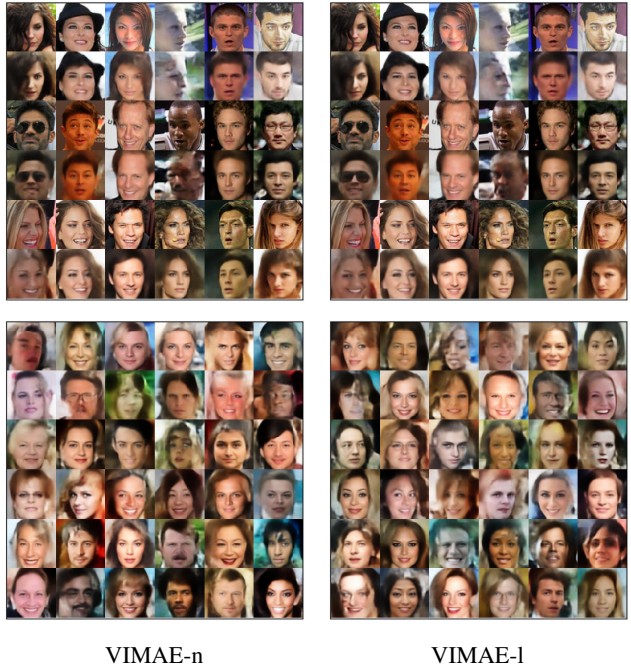

VIMAE-n                    VIMAE-l

Figure 4: Test reconstruction (top) and random samples (bottom) of the two VIMAE models, $\lambda = 10$.

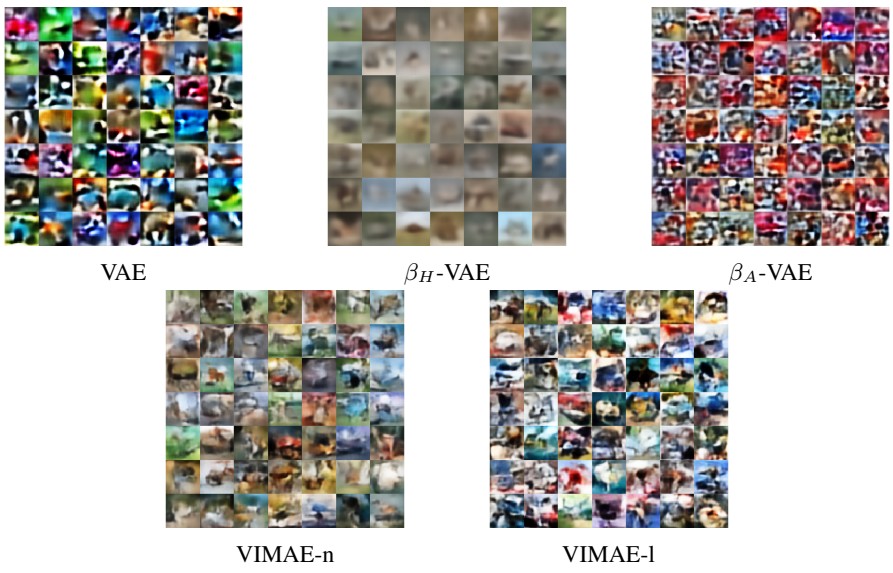

VAE                    $\beta_H$-VAE                    $\beta_A$-VAE

VIMAE-n                    VIMAE-l

Figure 5: random samples generated by the VAEs trained on CIFAR10

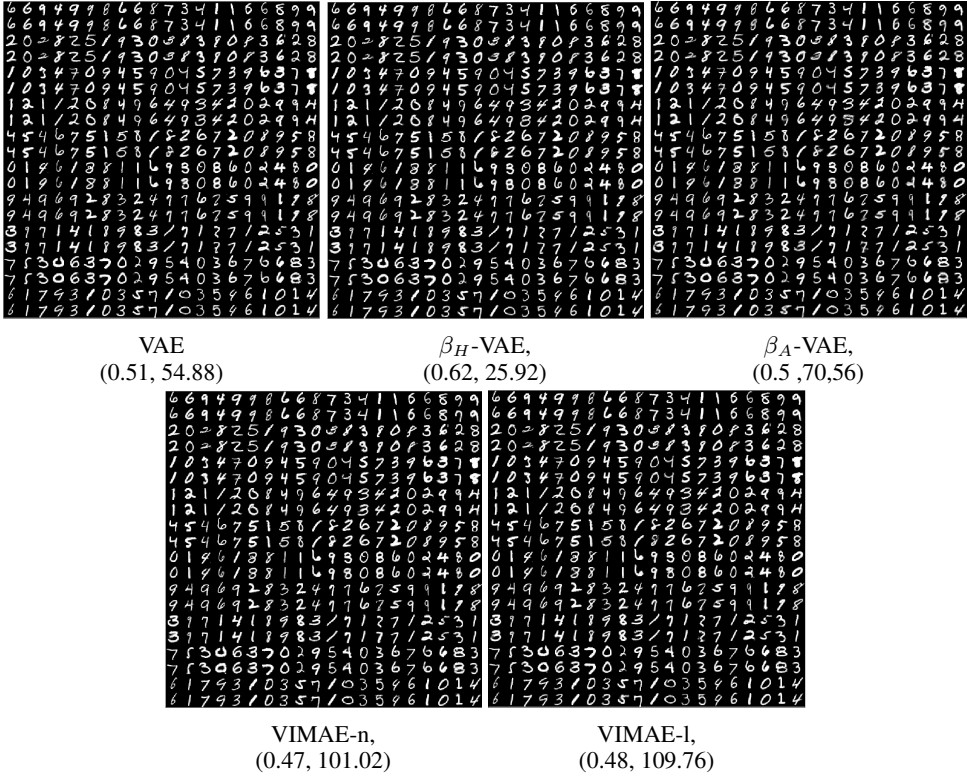

Figure 6: Test reconstruction by the VAEs trained on MNIST

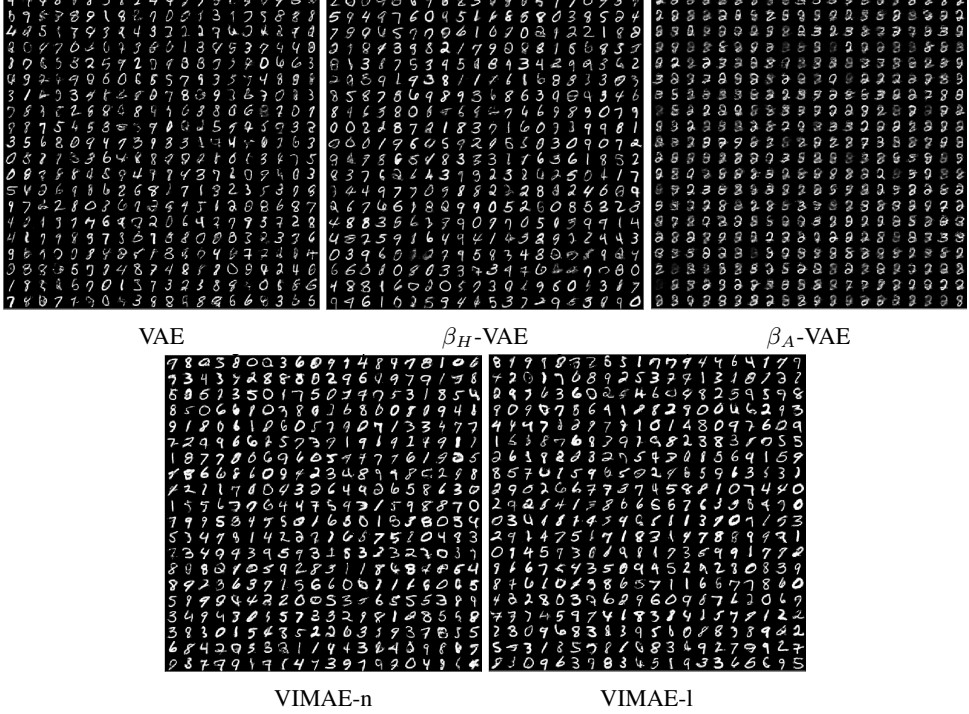

Figure 7: random samples generated by the VAEs trained on MNIST

