# OpenReview forum: "The Variational InfoMax AutoEncoder"
_ICLR.cc/2020/Conference — Reject_

### Official Review · AnonReviewer2 · 2019-10-15
**Official Blind Review #2**

**Rating:** 3

**Review:**

The paper develops an information-theoretic training scheme for Variational Auto-Encoders (VAEs). This scheme is tailored for addressing the well-known disentanglement problem of VAEs where an over-capacity encoder sometimes manages to both maximize data fit and shrink the KL divergence between the approximate posterior and prior to zero. Consequently, the latent representations of the observations become independent, making them unusable for any downstream task.

The method developed in Section 3 and proposed explicitly in Eqs 7 to 11 is novel per se, though not groundbreaking.

Figs 2 and 3 are only few examples manually chosen from rather simple tasks. In the absence of a quantitative evaluation metric, they are not informative. In the outcomes of the same runs, there might exist counterexamples where the vanilla VAE generate perceptively more appealing reconstructions than VIMAE.

As a minor comment, I think the paper unnecessarily complicates the presentation of the idea. The development of notions such as f-Divergence, links to MMD, the constrained optimization setting in Eq 5 etc., do not serve to the main story line, but only distracts a mind. I would prefer a brief intro on VAEs and directly jumping into the proposed method. Last but not least, the statement of technical novelty in Section 3 is extremely encrypted. What exactly is "absolutely" novel there and what is prior art? Can the authors give a to-the-point answer to this question during the rebuttal?

The paper has two fundamental weaknesses:
i) The paper misses a key reference, which addresses the same representation disentanglement problem using the same information-theoretic approach, only with some minor technical divergences:

Alemi et al., Fixing a Broken ELBO, ICML, 2018.

This paper is a must-cite, plus a key baseline. This submission cannot be treated as a contribution without showing an improvement on top of this extremely closely related work. The problem is the same, the solution is almost the same, and the theoretical implications of the solution are also the same.

ii) The paper builds the entire story line around the statement that the representation disentanglement problem is caused by the properties of the ELBO formula and attempts to fix it by developing a new inference technique. This is attitude largely overlooks simpler explanations. For instance, the vanilla VAE assumes a mean field q(z|x) across the z dimensions. This makes q(z|x) fall largely apart from p(x) which definitely does not factorize that way. Earlier work has shown substantial improvements on the quality of q(z|x) when structured variational inference techniques are used, such as normalizing flows. Furthermore, there has also emerged techniques that can very tightly approximate p(x) in closed-form using the change of variables formula without performing any variational inference at all. No need to say, a tight approximation on p(x) means the same for p(z|x) simply using the Bayes rule. This leaves no room for further improvements by information-theoretic inference alternatives. For instance see:

Dinh et al., Density Estimation Using Real NVP, ICLR, 2017

The presence of such two strong alternative remedies to the problem addressed by this work makes its fundamentals shaky. The only way to get over this situation is to provide a thorough comparison against these methods, which is obviously missing.

---
Post-rebuttal: Thanks to authors for all the effort they put on the updated version. I appreciate that the paper now provides a quantitative comparison. However, my point (ii) remains fully unaddressed and I fully share the theoretical concerns raised by Reviewer 1. All in all, I keep my position on the below the threshold side with a weak reject.

**Experience Assessment:**

I have published one or two papers in this area.

**Review Assessment: Checking Correctness Of Derivations And Theory:**

I carefully checked the derivations and theory.

**Review Assessment: Checking Correctness Of Experiments:**

I carefully checked the experiments.

**Review Assessment: Thoroughness In Paper Reading:**

I read the paper thoroughly.

---

> ### Author Response · Authors · 2019-11-15
> **Response to Reviewer #2**
>
> We thank the reviewer for the detailed review and for the thoughtful comments.
>
> > Figs 2 and 3 are only few examples manually chosen from rather simple tasks.
> In order to evaluate the quality of the generated samples we used the Frechet Inception Distance for the CIFAR10 and CelebA dataset (see the appendix) and the Negative Log Likelihood for the Omniglot, in all these cases the quality of the samples of VIMAE models is higher. In order to evaluate the reconstruction we consider the L_2 norm, and as suggested by (Alemi et al., 2017) each model is denoted by its reconstruction loss and rate term, helping us to observe that the VIMAE models are learning the most informative generator.
>
> >  I think the paper unnecessarily complicates the presentation of the idea.
> Following your suggestion in the revised version we moved the unnecessary details to the story in other sections, in order to do distract from the main story line
>
> We compare the VIMAE and the model proposed by (Alemi et al., 2017), observing that both are controlling the encoding information, and that the two coincide in the case the bound of the rate term is equal to the network capacity. We observed also that the main difference between the two approaches is in the ratio decoding information/capacity, in VIMAE is one for any p(z), in the Alemi model is in general less than one, that means, according to theory of separability described in (Mathieu et al., 2019) and confirmed by the experiments, that the learned representation is not in general robust and clustered.
> We observe moreover that the model as described in (Alemi et al, 2017) is not in general feasible to compute and it is necessary to consider a $\beta$-VAE with $\beta<1$.
>
> In the related work section we underlined that  in order to avoid the uninformative issue is possible to consider a family of models learning a flexible prior $p(z)$. But we do not consider them, because are quite expensive to compute and they are not suitable for tasks like the lifelong where different types of data are considered

---

### Official Review · AnonReviewer3 · 2019-10-28
**Official Blind Review #3**

**Rating:** 6

**Review:**

Overview: This paper describes the Variational InfoMax AutoEncoder (VIMAE), which is based on the learning principle of the Capacity Constrained InfoMax. The core idea behind VIMAE is that the encoding information is not bounded while network capacity is. The issue that VIMAE can handle, and where VAE fails, is that representations are not informative of input data, due to the information bottleneck idea that VAE is built upon. The authors describe InfoVAE and β-VAE, which both attempt to solve this problem. The theory behind VIMAE is then described and tested against VAE and β-VAE, in their abilities to evaluate the entropy of Z, in reconstruction and generative performance, and in robustness to noise and generalization.

Contributions: The authors clearly state the contributions of the paper themselves, but in a summary: a derivation of a variational lower bound for the max mutual info of a generative model, definitions and bounds estimation for a VAE, associations for generative quality and disentanglement representation to mutual information and network capacity, and finally proposing the Capacity-Constrained InfoMax.

Comments:
Page 1: “Our derivation allows us to define…” -> this sentence is a bit long and took me a few reads to understand, reword this please.
“Derivation of a variational lower bound for ...a geneaive* model belonging..” -> “generative”
Page 2: “We conclude the paper with experimental results and conclusions.” You already said conclude twice in this sentence, I feel like this could be better worded to avoid that.
Page 5: “In order to test the assumption that it is sufficient..” -> This wording is also hard to wrap my head around. Too many commas.
Page 6: “In figure 1 are plotted the 2d…” -> This sentence could be reworded

My main complaint with the paper is that there are quite a few places that could be reworded better in addition to these. Please fix this.

You mention both InfoVAE and β-VAE yet only test your models against a β-VAE model. What was your reasoning for this?

The semi-supervised learning experiment is interesting (the one based on Zhao et al.) VMAE models are able to classify better regardless of noise or sampling procedure, especially the smaller capacity model VMAE-1. I’d like to hear a discussion in the paper about future work and how the authors believe this could be applied elsewhere.


**Experience Assessment:**

I do not know much about this area.

**Review Assessment: Checking Correctness Of Derivations And Theory:**

I did not assess the derivations or theory.

**Review Assessment: Checking Correctness Of Experiments:**

I assessed the sensibility of the experiments.

**Review Assessment: Thoroughness In Paper Reading:**

I read the paper at least twice and used my best judgement in assessing the paper.

---

> ### Author Response · Authors · 2019-11-15
> **Response to Reviewer #3**
>
> We thank the reviewer for the detailed review and for the thoughtful comments.
>
> In the revised version we reworded the critical parts and we underlined that the VIMAE, in the case the prior $p(z)$ is normally distributed coincides with the InfoVAE (Zhao et al., 2017) and WAE (Tolstikhin et al., 2017).
>
> In the conclusion we suggest a future application to lifelong learning; indeed,the ability to learn the relevant features of the data in a robust way is suitable and particularly important in a lifelong task, where it is necessary to learn only few specific features for each task (dataset) in order to avoid the catastrophic forgetting issue.

---

### Official Review · AnonReviewer1 · 2019-10-31
**Official Blind Review #1**

**Rating:** 1

**Review:**

I went over this work multiple times and had a really hard time judging the novelty of this work. The paper seems to be a summary of existing work reinterpreting variational autoencoding objectives from an information theoretic standpoint. In particular, the paper seems to follow the same analysis as in Wasserstein Autoencoders (Tolstikhin et al., 2017) and InfoVAE (Zhao et al., 2017).  It is unfair to say that the objectives were "derived independently" since these works are from a couple of years ago.

The paper also lacks discussion on two crucial works in this space:
1. https://arxiv.org/abs/1711.00464 shows how to trade off rate-distortion in VAEs.
2. https://arxiv.org/abs/1812.10539 shows how to learn informative representations by eliminating the KL divergence term and at the same time specifying an implicit generative model (Theorem 1).

re: disentanglement. Unsupervised disentanglement has been shown to be theoretically impossible and several key challenges have been highlight w.r.t. prior work. Again, the relevant paper: https://arxiv.org/abs/1811.12359 has not even been cited. More importantly, the claims around "more disentangled representation" are imprecise in light of this work.

A proper discussion on the contributions of this work as well as discussion on the above related works would be desirable on the author's end.

**Experience Assessment:**

I have published one or two papers in this area.

**Review Assessment: Checking Correctness Of Derivations And Theory:**

I assessed the sensibility of the derivations and theory.

**Review Assessment: Checking Correctness Of Experiments:**

I carefully checked the experiments.

**Review Assessment: Thoroughness In Paper Reading:**

I read the paper thoroughly.

---

> ### Author Response · Authors · 2019-11-15
> **Response to Reviewer #1**
>
> We thank the reviewer for the detailed review and for the thoughtful comments.
>
> In the revised version we underline that the main contribution of the paper is not the definition of a model, that was previously defined in  (Tolstikhin et al., 2017) and (Zhao et al., 2017),  but the information analysis of such model. Thanks to this description we was able to describe the relationship between the decoding information and the network capacity, and then the introduction of the Capacity Constrained InfoMax.
>
> In this new version we cite both (Alemi, et al., 2017) and (Grover and Ermon, 2019).
> We observed that the model introduced in (Grover and Ermon, 2019) belongs in the  (unbounded) InfoMax family, but we do not compare it with the other models since the sampling is quite expensive, and then out of our interest.
> Instead we compare both theoretically and computationally the VIMAE model with the one proposed in (Alemi, et al., 2017), observing that if in some special cases the two approaches are identical (minimal bound = Network capacity), computationally the VIM objective is easier to compute, and it is learning more robust and clustered representation.
>
> Since the word disentangled was not clear and incorrect in this context,  in the revised version we try to describe  what, in this paper, is a good representation: a robust and clustered in order to be used as input for simple classifier. This definition of good is similar to what in (Mathieu, et al., 2019) is defined decomposed.

---

### Decision · Program_Chairs · 2019-12-19

**Decision:**

Reject

**Comment:**

This paper describes a new generative model based on the information theoretic principles for better representation learning. The approach is theoretically related to the InfoVAE and beta-VAE work, and is contrasted to vanilla VAEs. The reviewers have expressed strong concerns about the novelty of this work. Some of the very closely related baselines (e.g. Zhao et al., Chen et al., Alemi et a) are not compared against, and the contributions of this work over the baselines are not clearly discussed. Furthermore, the experimental section could be made stronger with more quantitative metrics. For these reasons I recommend rejection.